# Circulating microRNA miR-425-5p Associated with Brain White Matter Lesions and Inflammatory Processes

**DOI:** 10.3390/ijms25020887

**Published:** 2024-01-10

**Authors:** Sandra Van der Auwera, Sabine Ameling, Katharina Wittfeld, Stefan Frenzel, Robin Bülow, Matthias Nauck, Henry Völzke, Uwe Völker, Hans J. Grabe

**Affiliations:** 1Department of Psychiatry and Psychotherapy, University Medicine Greifswald, 17475 Greifswald, Germany; 2German Centre for Neurodegenerative Diseases (DZNE), Site Rostock/Greifswald, 17475 Greifswald, Germany; 3Interfaculty Institute for Genetics and Functional Genomics, University Medicine Greifswald, 17475 Greifswald, Germany; 4German Centre for Cardiovascular Research (DZHK), Partner Site Greifswald, 17475 Greifswald, Germany; matthias.nauck@med.uni-greifswald.de (M.N.);; 5Institute for Diagnostic Radiology and Neuroradiology, University Medicine Greifswald, 17475 Greifswald, Germany; 6Institute of Clinical Chemistry and Laboratory Medicine, University Medicine Greifswald, 17475 Greifswald, Germany; 7Institute for Community Medicine, University Medicine Greifswald, 17475 Greifswald, Germany

**Keywords:** circulating miRNAs, vascular dementia, white matter lesions, inflammatory processes, general population, *SH3PXD2A*, miR-425-5p, moderation analysis

## Abstract

White matter lesions (WML) emerge as a consequence of vascular injuries in the brain. While they are commonly observed in aging, associations have been established with neurodegenerative and neurological disorders such as dementia or stroke. Despite substantial research efforts, biological mechanisms are incomplete and biomarkers indicating WMLs are lacking. Utilizing data from the population-based *Study of Health in Pomerania* (SHIP), our objective was to identify plasma-circulating micro-RNAs (miRNAs) associated with WMLs, thus providing a foundation for a comprehensive biological model and further research. In linear regression models, direct association and moderating factors were analyzed. In 648 individuals, we identified *hsa*-miR-425-5p as directly associated with WMLs. In subsequent analyses, *hsa*-miR-425-5p was found to regulate various genes associated with WMLs with particular emphasis on the *SH3PXD2A* gene. Furthermore, miR-425-5p was found to be involved in immunological processes. In addition, noteworthy miRNAs associated with WMLs were identified, primarily moderated by the factors of sex or smoking status. All identified miRNAs exhibited a strong over-representation in neurodegenerative and neurological diseases. We introduced *hsa*-miR-425-5p as a promising candidate in WML research probably involved in immunological processes. Mir-425-5p holds the potential as a biomarker of WMLs, shedding light on potential mechanisms and pathways in vascular dementia.

## 1. Introduction

The term dementia covers a broad spectrum of progressive neurodegenerative conditions, notably Alzheimer’s disease (AD) and vascular dementia (VD), characterized by a gradual decline in cognitive function, memory impairment, and behavioral changes [1]. Despite our current understanding, the exact etiology and mechanisms of various dementia subtypes remain elusive. However, extensive research has identified several risk factors associated with their development, including genetic predisposition, advanced age, and lifestyle [2]. While the hallmark pathology of AD involves the accumulation of beta-amyloid plaques and tau-tangles in the brain [3], VD is mainly characterized by impaired cerebral blood flow, and emerging evidence suggests that stroke and white matter lesions (WML) significantly contribute to the onset and progression of VD [4,5]. WMLs refer to abnormal areas of increased signal intensity, typically observed in brain white matter, as depicted in T2-weighted magnetic resonance imaging (MRI) scans [6,7]. These lesions are commonly encountered in the aging population, even without clinical pathology, but have been recognized as risk factors for various neurological conditions [6]. Moreover, WMLs have been associated with other pathological processes implicated in dementia, such as inflammation, oxidative stress, and amyloid-beta plaques [5]. These factors may interact synergistically, creating a neurodegenerative environment that accelerates the onset and progression of dementia.

Several risk factors for the higher burden of WMLs and associated dementias are known. Among those, one key variable investigated is the influence of sex. Studies have shown that women tend to exhibit a higher burden of WMLs compared to men, even when considering age-related factors, possibly due to underlying hormonal and genetic factors [8] or different comorbidity patterns [9]. Lifestyle factors such as smoking have also been implicated in the development and progression of WMLs, as smoking has a strong impact on vascular diseases, including cerebral small vessel disease or stroke [9,10]. Additionally, evidence suggests a substantial genetic component driving the occurrence of WMLs with the *APOE4 (Apolipoprotein E)* genetic locus, a key genetic risk factor for late-onset AD, as a common genetic factor driving both WML and AD [11,12].

Emerging research has focused on the involvement of micro-RNAs (miRNAs) in the development and progression of dementias and neurodegenerative processes [13,14]. MiRNAs are small non-coding RNA molecules that play crucial roles in post-transcriptional regulation and gene expression. As miRNAs are released from cells into biofluids and tend to be very stable, they serve as promising biomarkers for complex traits. Studies have identified specific miRNAs that are dysregulated in AD patients, suggesting their potential as diagnostic markers for early detection and disease monitoring [15]. Until now, very few studies have investigated the link between miRNAs and WMLs in the context of neurodegeneration in only small sets of participants [16,17]. Nevertheless, certain miRNAs have been found to regulate genes involved in white matter integrity and vascular function, suggesting a mechanistic link between miRNAs, WMLs, and dementia pathology [11]. Investigating this link is of increasing importance as reliable biomarkers for early detection of neurodegenerative processes leading to dementia are still lacking. MiRNAs could also give mechanistic insights into biological processes implicated in dementia. Analyzing individuals from the general population who are at higher risk for WMLs is of increasing importance to identify biological causes and mechanisms in a prodromal phase of dementia.

In this study, we aim to investigate the association between 171 plasma-circulating miRNAs and WMLs in a cohort comprising 648 individuals from the general population. Moreover, we will explore the influence of potential moderating factors and put our findings into the context of other neurodegenerative diseases and associated biological mechanisms.

We hypothesize that identified miRNAs show associations with biological pathways related to dementia so far. These insights will introduce new candidates for WML research in the general population and introduce promising targets for further research on disease prevention and biomarkers.

## 2. Results

### 2.1. Clinical Impact of WML

Participant characteristics of the TREND-0 plasma-circulating miRNA and MRI sample are given in Table 1. After quality control and excluding subjects with missing covariate data, the MRI sample included data from 1854 subjects, 648 of them with additional data on plasma-circulating miRNAs. With regard to the three parametrizations of WMLs (prevalence, number of lesions, and total lesion volume), in both subsamples, the prevalence of WMLs was above 70% with the number of lesions between 0 and 37, and the total lesion volume between 0 and 44 cm^3^. The relationship between total WML volume and the number of lesions in both subsamples was nonlinear. Specifically, the total volume of WMLs showed a linear increase with the number of WMLs, but only until reaching a plateau (Appendix A), which could reflect the conjunction of smaller lesions into one larger lesion.

To identify the phenotypic implications of WMLs, we tested for associations between WMLs and memory impairment. WMLs in both subsamples were associated with cognitive parameters (Appendix A). A higher burden of WMLs was linked to a general decline in verbal memory performance. Notably, these effects were more pronounced for immediate recall, particularly in the miRNA sample. Comparing the different WML parameters, significant effects on memory performance were observed for all three parametrizations.

These findings underscore the necessity to investigate and interpret the underlying biological mechanism affecting total WML volume and the number of WMLs separately and confirm that the TREND-0 data reflect reasonable phenotypical associations.

### 2.2. hsa-miR-425-5p Is Associated with WMLs

To identify target circulating miRNAs affecting WMLs, direct associations for 171 miRNAs were performed. We identified five miRNAs showing nominal significance for total WML volume, seven for the number of WMLs, and 13 for the presence of WMLs (Figure 1A, Appendix A). Of those, only *hsa*-miR-425-5p survived multiple testing corrections (*p_BH_* = 0.01, β = −0.83; Table 2) for the presence of WMLs. Higher levels of *hsa*-miR-425-5p were associated with higher prevalence for WMLs and greater overall WML burden. The *MIR425* gene is located on chromosome 3 within the *DALRD3 (DALR anticodon-binding domain-containing protein 3)* gene. Both *hsa*-miR-425-5p and the *MIR425* and *DALRD3* genes are expressed in the human brain (Appendix A). Data from mouse central nervous system (CNS) samples indicate that miR-425-5p expression is consistently enriched in microglia and immune cells within the nervous system (Figure 1B and Appendix A) and highly conserved across vertebrates. In humans, the *DALRD3* gene region has previously been associated with various brain-related traits (Appendix A), educational attainment, and coronary artery disease (Appendix A).

Of all the miRNAs tested, only *hsa*-miR-152-3p reached nominal significance towards all three WML parametrizations. The corresponding miRNA gene is located on chromosome 17 within the *COPZ2 (COPI coat complex subunit zeta 2)* gene. The miRNA and *COPZ2* genes are expressed in the human brain (Appendix A). Previously, this gene locus was associated with intelligence, educational attainment, Parkinson’s disease, and AD (Appendix A). In mouse CNS, miR-152-3p is mainly enriched in astrocytes and is also broadly conserved (Appendix A).

#### 2.2.1. Association of Target miRNAs with Structural AD-Related MRI Phenotypes

Subsequent analyses revealed that both miRNAs, *hsa*-miR-425-5p and *hsa*-miR-152-3p, exhibited no significant association with total white matter volume in general. There was also no significant association between these two miRNAs and structural MRI-based measures related to AD, such as total hippocampal volume or AD score (all *p*-values > 0.45). These findings underscore that the identified target miRNAs show specific effects on WMLs as vascular brain parameters and exhibit no overall general neurodegenerative effect on brain parameters.

#### 2.2.2. The Role of *hsa*-miR-425-5p in Inflammation

As previous research suggests a prominent role of inflammation in the development of neurological diseases [18,19] and involvement of *hsa*-miR-425-5p in microglia inflammation in a mouse stroke model [20], we examined the association between *hsa*-miR-425-5p and the blood-based inflammation markers CRP and fibrinogen. Both inflammation markers were negatively associated with the abundance of *hsa*-miR-425-5p (*p*_CRP_ = 0.026, *p*_FIB_ = 0.001). There was no significant association between the inflammatory markers and the WML parameters. This suggests that the regulation of peripheral inflammatory processes might be a possible candidate mechanism.

#### 2.2.3. Target Genes of *hsa*-miR-425-5p in WMLs GWAS

To identify miRNA target genes implicated in WML biology, we compared possible target genes of *hsa*-miR-425-5p with GWAS summary statistics on WMLs. We used publicly available databases where we identified 620 unique target genes for *hsa*-miR-425-5p of which 582 were left for analysis (38 on chromosome X excluded). For the WML GWAS, we retrieved 53 SNPs in the target gene *SH3PXD2A (SH3 and PX domains 2A)* that reached genome-wide significance (Figure 1D) (additional four genes *CTSS* (*cathepsin S*), *EPN2* (*epsin 2*), *RIT1* (*Ras-like without CAAX 1*), and *PPP4R3A* (*protein phosphatase 4 regulatory subunit 3A*) with SNPs *p* < 5 × 10^−6^ (Appendix A)). These genes regulate pathways such as signaling pathways (Rho and RAC1 GTPase, p38 MAPKinase), immune system, neuronal pathways (neurotoxicity, neuronal development, and differentiation), or cellular transport. The top gene *SH3PXD2A* is generally expressed in the brain (Appendix A) and especially in white matter (Figure 1C and Appendix A). Previous associations are known with white matter volume, stroke, depression, educational attainment, blood pressure, and cardiological markers, demonstrating a connection between cardiovascular and cerebrovascular endpoints. According to the Human Protein Atlas, *SH3PXD2A* belongs to a gene cluster of nonspecific immune responses in the brain and is directly associated with the *ADAM15 (ADAM metallopeptidase domain 15)* gene.

### 2.3. Influence of Moderating Factors

*Moderation by sex:* In sex interaction models, 19 and 18 nominally significant miRNAs were identified for total WML volume and number of WMLs, respectively (Appendix A, Appendix A). Only for the number of WMLs, two miRNAs, *hsa*-miR-126-3p and *hsa*-miR-374a-5p, survived multiple tests (Table 2). In both cases, lower values of miRNA abundance (reflected in higher ΔCt values) were associated with a higher WML burden in females but with a reduced burden in males (Appendix A). Both miRNAs are expressed in the brain (Appendix A) and the *EGFL7 (EGF-like domain multiple 7)* gene harboring *MIR126* on chromosome 9 has been associated with AD as well as white matter growth (Appendix A).*Moderation by APOE ε4:* In interaction analyses with the *APOE ε4* carrier status, 13 and 19 nominally significant miRNAs were identified for total WML volume and number of WMLs, respectively (Appendix A, Appendix A). None of them survived multiple testing corrections. The lowest *p*-value was observed for *hsa*-miR-140-5p on the number of WMLs (*p* = 0.0011). Carriers of the *APOE ε4* allele had a beneficial outcome for WML burden in the case of high levels of *hsa*-miR-140-5p, whereas in the case of lower levels, the WML burden increased (Appendix A). *Hsa*-miR-140-5p is only slightly expressed in the brain (Appendix A) and its harboring gene *WWP2 (WW domain containing E3 ubiquitin protein ligase 2)* has been associated with addictive behavior and cognitive traits (Appendix A).*Moderation by smoking status:* Interaction with smoking status revealed 13 and 15 nominal significant miRNAs for total WML volume and number of WMLs, respectively (Appendix A, Appendix A). Three miRNAs reached BH-corrected significance in the latter model (*hsa*-miR-885-5p, *hsa*-miR-199a-5p, *hsa*-miR-194-5p; Table 3, Appendix A) with *hsa*-miR-199a-5p reaching significance in both WML models. Interestingly, only *hsa*-miR-885-5p shows a substantial expression in brain tissues (Appendix A) and the strongest link towards neurodegenerative endpoints concerning its harboring gene *ATP2B2 (ATPase plasma membrane Ca2+ transporting 2)* (Appendix A).

### 2.4. Significant Plasma-Circulating miRNAs Are Enriched in Neurodegeneration

Taking the set of six significant plasma-circulating miRNAs with *p_BH_* < 0.1 from all analyses (Table 2), we investigate their over-representation in neurodegenerative diseases. The analyses confirmed, among others, a significant over-representation of neurodegenerative or nervous system diseases in general (Table 3). For AD and neurodegenerative and vascular diseases, all six miRNAs were implicated. The results for all nominal significant miRNAs from all individual analyses (direct and interaction analyses) are shown in the Appendix A with, again, strong enrichment for neurodegenerative and neurological diseases including AD and stroke. This shows the impact of miRNAs associated with neurodegenerative traits in the context of WMLs even in the apparently healthy general population.

## 3. Discussion

We identified candidate plasma-circulating miRNAs mainly associated with the quantity and presence of WMLs in the general population showing a strong over-representation in neurodegeneration. The identified miRNAs showed associations toward neurodegenerative endpoints and immune-related processes, even within this apparently healthy general population. The results suggest that biological changes attributable to the burden of WMLs may have implications on the biological manifestation of subsequent neurodegenerative diseases.

On the clinical level, we found that a higher burden of WMLs was associated with lower cognitive performance, even after controlling for health-related factors. These findings are consistent with previous research [21], highlighting that cognitive changes can be observed at a prodromal stage of disease. Stronger effects were generally found for the immediate recall of words. This pattern of cognitive changes aligns with typical cognitive changes observed in VD [22], where the early-phase immediate recall, involving the reception of information, is affected, while the hippocampal mechanism for storing and recalling information after a longer time remains relatively unimpaired.

Analyzing the impact of plasma-circulating miRNAs on WMLs introduced *hsa*-miR-425-5p as a promising candidate not only directly associated with WMLs but also targeting genes identified in large GWAS analyses for WMLs. Prior studies have linked miR-425-5p to AD and neurodegeneration (Table 4) and suggested its role as a key miRNA regulating important AD-associated genes [23]. In line with these results, our study demonstrated that higher levels of this miRNA were associated with a higher incidence of WMLs in general and a greater overall number of WMLs, which aligns with previous results on AD [24,25]. Interestingly, no miRNA was significantly associated with total WML volume, the most commonly analyzed endpoint in WML analyses [11]. Even *hsa*-miR-425-5p showed no association. This suggests that *hsa*-miR-425-5p might play a role in the initiation and onset of the pathology but probably not in lesion size and volume. Thus, our findings suggest that the number of WMLs in addition to the total volume may hold additional biological information, as both parameters were also nonlinearly related in our sample. Moreover, the results for common structural MRI-based AD markers revealed no significant association, suggesting that *has*-miR-425-5p might be specific to induce new lesions and has no overall degenerative effect on brain structure. Analyzing WML GWAS data, we identified several genetic variants in target genes has*hsa*-miR-425-5p significantly associated with WMLs, particularly drawing attention to the gene *SH3PXD2A*, which is highly expressed in brain white matter and involved in core mechanisms of neurodegeneration, such as blood–brain barrier dysfunction, immune response, and amyloid-beta neurotoxicity [26,27]. Furthermore, *SH3PXD2A* is directly interacting with the *ADAM15* gene involved in neurodegeneration and inflammatory processes [28]. Previously reported associations of *hsa*-miR-425-5p with immunological processes [20] were indicated in our data concerning blood-based markers (CRP, fibrinogen), but a potential association of these markers with WMLs was not observed. This suggests that the immunological mechanisms underlying *hsa*-miR-425-5p might be different in the brain and the peripheral system and be based on a more complex biological moderation. In addition, mouse data implicated a potential role of mir-425-5p in microglia activation. Another significant target, *hsa*-miR-152-3p, was associated with all three WML endpoints, at least on a nominal significance level. Previous studies have shown that overexpression of *hsa*-miR-152-3p has an alleviating effect on neuronal degeneration and apoptosis in VD, which aligns with our results on WMLs [29]. It has also been identified as a target for treatment with tilianin, which demonstrated neuroprotective effects in rat models of VD [30,31]. Thus, our results confirm that biological informative results can be obtained implicating an involvement of specific miRNAs in WMLs in this healthy general population. These results could be the first step to uncover the regulatory role of miRNAs in WMLs and VD and should be investigated in further mechanistic analyses.

In addition, we investigated the influence of biological and behavioral moderating factors previously linked to WMLs on the association between plasma-circulating miRNAs and WMLs. Two miRNAs, *hsa*-miR-126-3p and *hsa*-miR-374a-5p, demonstrated a strong sex dependency, and, especially, *has*-miR-126-3p has previously been associated with AD and related traits, such as neuroinflammation, neurogenesis, or BDNF synthesis (Table 4) but with contrary directions of effect. This disparity may, in part, be attributed to sex effects, underscoring the importance of conducting sex-specific analyses in identifying biological disease mechanisms [32,33]. On the other hand, *hsa*-miR-374a-5p indicated a stronger link to cardiovascular phenotypes, which are general risk factors for WML and VD with different prevalences in males and females [2,9]. Regarding smoking behavior, three miRNAs, *hsa*-miR-885-5p, *hsa*-miR-199a-5p, and *hsa*-miR-194-5p, were identified which have been associated with AD, as well as metabolic risk profiles (Table 4), potentially linking smoking as a cardiovascular risk factor with alterations in brain white matter and neurodegeneration. Interestingly, moderation analysis on the genetic *APOE ε4* status did not reveal a significant miRNA association. Nevertheless, the top hit *hsa*-miR-140-5p has been found to exert neuroprotective effects and has been associated with AD previously (Table 4). The exploration of possible connections of the set of six significant target miRNAs with other diseases revealed a strong over-representation of neurodegenerative, neurological, and psychiatric diseases. These findings provide further insights into the potential involvement of miRNAs in the pathogenesis of WMLs and related neurodegenerative processes.

Taken together, the strengths of the current study encompass several key aspects. Firstly, we distinguished between different WML endpoints in miRNA associations, providing a comprehensive analysis of their specific relationships. Additionally, we included important risk factors in moderation analyses, which allowed us to explore how these factors may influence the miRNA effects. Furthermore, we considered additional MRI endpoints and immunological markers to analyze potential mechanisms involving *hsa*-miR-425-5p. However, our study was limited by the lack of additional longitudinal or clinical disease samples to further explore in our results. We also worked with a panel of only 171 preselected miRNAs, capturing just a small amount of miRNAs detectable in plasma, so we could have missed relevant biological signatures. To prove the hypothesis of *hsa*-miR-425-5p as a suitable biomarker for WML, additional clinical studies involving dementia patients are needed. In addition, functional studies need to be performed to identify the implicated biological mechanisms of this target miRNA and derive regulatory models that could also drive drug discovery. Despite these facts, our study demonstrates that valuable results can be obtained from general population samples, where biological markers can be identified before the onset of diseases. This aspect is particularly relevant, as the identification of reliable biomarkers remains a primary objective in the research on AD and related dementias [34]. Further research should investigate the biological mechanisms including *hsa*-miR-425-5p and their target genes in neurodegeneration.

**Table 4 ijms-25-00887-t004:** Previous associations of significant miRNAs with neurodegenerative traits found in Pubmed (selection).

miRNA	Previous Results Regarding AD and/or Cognition from Pubmed
***hsa*-miR-425-5p**	Regulation of AD pathogenic genes [23] Interacting with BACE1 [35] Upregulated in AD [24,25] Association with memory and learning disorders [36] Promotes formation of Aβ plaques [37]
***hsa*-miR-126-3p**	Overexpression could reduce Aβ plaque area and neuroinflammation in the hippocampus [38] Associated with inflammation in the pathogenesis of AD [39] Involved in neurogenesis [40] Upregulated in plasma of AD patients [41] Altered regulation in brain of AD male rats [42] Involved in neuronal accumulation of AD [43] Decreased in plasma of AD subjects [44] Part of a nine-miRNA signature as potential biomarker for AD [45] Dysregulated in plasma of AMD rats [46] Associated with stroke recovery [47] Negative correlation with cognitive function [48] Dysregulated in AD NMV [49] Cardiovascular events (including stroke) [50] Regulation of BDNF synthesis [51]
***hsa*-miR-374a-5p**	Part of plasma signature of obstructive sleep apnea in AD [52] Cardiovascular events (including stroke) [50] Overexpression reduces cell apoptosis [53]
***hsa*-miR-885-5p**	Regulating neuronal cell injury [54] Serum biomarker for AD [55] Upregulated in AD [56] Associated with higher metabolic risk profile in older subjects [57]
***hsa*-miR-199a-5p**	Involved in AD development [58] Related to cognitive impairment [59] Link between AD and diabetes [60] Protects cognitive function in ischemic stroke [61]
***hsa*-miR-194-5p**	Association with WML and cognitive impairment [16] Associated with higher metabolic risk profile in older subjects [57] Downregulated in blood of AD patients [62] Inhibit apoptosis of hippocampal neurons [63]
***hsa*-miR-140-5p**	Risk factor for memory impairment induced by Aβ [64] Associated with neurodegenerative diseases in general [65] Associated with vascular cognitive impairment [66] Associated with cognitive performance in healthy older adults [67] Associated with AD risk gene ADAM10 [68] Neuroprotective effects [69]

In this paper, we provide the first study with a comprehensive association analysis between miRNA alterations and WMLs in a large general population sample. We propose *hsa*-miR-425-5p as a promising candidate probably involved in inflammatory mechanisms. Additionally, we identified several plasma-circulating miRNAs associated with WMLs that also revealed a strong link towards neurodegeneration and were moderated by important dementia risk factors. The results imply that WML-associated miRNAs can be detected in the general population before the onset of the disease. The introduced candidates should be investigated regarding their potential as early indicators or precursors of neurodegeneration, shedding light on potential mechanisms and pathways in WMLs and VD.

## 4. Materials and Methods

### 4.1. SHIP Sample

The investigations in the Study of Health in Pomerania (SHIP) were carried out in accordance with the Declaration of Helsinki, including written informed consent from all participants. The survey and study methods were approved by the institutional review boards of the University of Greifswald.

SHIP is a population-based study from the northeast of Germany [70] with the aim to assess the prevalence and incidence of common diseases and their risk factors in the population. From 2008 to 2012, the SHIP-TREND-0 sample (hereafter named TREND-0) was recruited, including 4420 participants who underwent a standardized computer-assisted personal interview, during which they provided information on sociodemographic and lifestyle factors and also gave different biofluids for OMICS analyses. For 2047 participants, structural MRI data of the brain are available, and in a subsample of 708 TREND-0 participants, there are also data on plasma-circulating miRNA levels.

### 4.2. Verbal Memory Scores

The word list of the Nuremberg Age Inventory (NAI) was used as a measure for immediate and delayed verbal memory performance. The NAI is a German test developed to measure cognitive abilities during brain aging [71,72]. More details are given in the Appendix A. The resulting scores can be interpreted as a measure of short- and long-term memory capacity.

### 4.3. Brain Imaging Data

In addition, TREND-0 participants were asked for a whole-body MRI assessment. After the exclusion of subjects who refused participation or fulfilled exclusion criteria for MRI (e.g., cardiac pacemaker), 2047 subjects underwent the MRI scanning (for details of parameters and WML segmentation, see Appendix A). Quality control was performed for technical and medical parameters excluding motion artifacts, radiological findings, and known medical diagnoses for stroke, epilepsy, MS, Parkinson’s disease, or dementia. For analyses, we selected three different parameters characterizing WML burden: (1) Total lesion volume: total volume of all WMLs; (2) Number of lesions: counted number of all lesions detected; (3) Presence of any WML: coded as 0/1. For structural MRI measures related to AD and neurodegeneration, we selected total hippocampal volume as well as a structural AD score. Hippocampal volume was generated using FreeSurfer version 7.1.1 [73]. The structural MRI-based AD score was based on Freesurfer-generated features for cortical thickness, volumes of subcortical gray matter, white matter, and ventricles, as described elsewhere [74].

### 4.4. Plasma-Circulating miRNAs

Circulating miRNA levels from plasma were available in a subsample of 708 TREND-0 participants and measured in two distinct batches (371 and 337 subjects). In each batch, the influence of technical parameters was considered by the application of synthetic spike-ins, such as UniSp2, UniSp4, and UniSp5, which were added before the extraction of circulating plasma miRNAs. Before using RNA samples for miRNA profiling, the presence of spike-ins (UniSp2, UniSp4, UniSp5), yield of typical plasma miRNAs, absence of PCR inhibitors (UniSp6, Cel-miR-39, UniSp3), as well as hemolysis in the samples, was assessed by use of a microRNA QC PCR Panel V1.M (Qiagen, Hilden, Germany). Samples that did not pass the quality control were excluded from further processing. For RT-qPCR-based miRNA analysis, the Serum/Plasma Focus microRNA PCR-Panel (Qiagen, Hilden, Germany) V3.M and V4.M were used, covering 179 miRNAs. After quality control, technical parameters were regressed out of the data. The resulting residuals were used as independent variables in subsequent analyses. A batch was included in the analysis for a specific miRNA if at least 100 subjects contained a valid measurement of the respective miRNA (see Appendix A for a list of all miRNAs used), which resulted in 171 miRNAs for analysis. Further methodological details on miRNA preprocessing can be found in the Appendix A.

### 4.5. Immunological Markers

High-sensitivity C-reactive protein (hs-CRP) concentrations were determined in serum by nephelometry on the Dimension VISTA (Siemens Healthcare Diagnostics, Eschborn, Germany). Plasma fibrinogen concentrations were measured using the Clauss method assessed by coagulation analyzers (BCS-XP; Siemens Healthcare Diagnostics, Germany).

### 4.6. Additional Variables

As additional variables, current depression, educational attainment, smoking status, BMI, HbA1c, hypertension, hematocrit (HCT), platelet count (PLT), and *APOE ε4* status were chosen. For details on genetic data in SHIP, see the Appendix A.

### 4.7. Statistical Analyses

Subject characteristics of the sample were assessed by the mean, standard deviation, and range for metric variables and by numbers and percentages for categorical data. Different regression models were performed to assess the associations between circulating miRNAs and verbal memory performance on WMLs in the general population. Models for WMLs were performed for the metric variables total WML volume and number of WMLs and the binary variable presence of WMLs. For the two former ones, values were log-transformed prior to analysis (log_2_(1 + var)). For the binary trait, a binomial link was assumed in the GLM in case WMLs were used as the outcome. Models were calculated on the sample with non-missing data. For the MRI sample, this included all available data after quality control and non-missing data on memory scores, smoking, education, depression, *APOE* status, BMI, and hypertension, resulting in a final analysis sample of 1854 subjects. For the miRNA/MRI sample, this included all available MRI data after quality control as well as non-missing data for education, *APOE* ε*4* status, HCT, PLT, and hypertension, resulting in a final analysis sample of 648 subjects.

*Clinical impact of WMLs:* The following generalized linear models (GLM) were calculated with WMLs as predictors and verbal memory scores as outcomes.

Memory ~ WMLs + age + sex + smoking + education + depression + *APOE ε4* + BMI + hypertension + ICV

2.*Impact of miRNAs on WMLs:* GLMs were calculated with miRNA levels as predictors and WMLs (structural MRI markers) as outcome.

WMLs ~ miRNA + age + sex + education + *APOE ε4* + HCT + PLT + hypertension + ICV + batch

3.*Impact of target miRNAs and inflammatory markers:* In GLMs, the association between circulating inflammatory markers (CRP, fibrinogen) and significant miRNAs was investigated and corrected for age, sex, miRNA batch, smoking, BMI, education, HCT, and PLT. CRP was log-transformed prior to analyses.4.*Moderation effect of sex, APOE ε4, smoking:* Similar models as in 2. were performed, additionally including an interaction term between miRNA levels and sex, APOE ε4, or ever smoking.

All reported analyses were performed with R version 3.6.3. The interaction models were only performed for the metric variables of WML volume and number of WMLs due to small sample sizes in the intersection groups in the logistic models. Within the individual models 2. and 4., Benjamini–Hochberg correction for multiple testing was applied. In all analyses, age was taken nonlinearly into the model as restricted cubic splines. For the significant models, the normal distribution of residuals was investigated manually.

### 4.8. Post Hoc In Silico Analysis

In order to evaluate the impact of significant miRNAs on biological processes, we used publicly available databases and tools.

GWAS catalog [75], PhenoScanner v2 [76], Oxford BIG Server [77], and MiRNASNPv3 [78] were used to detect previous associations between SNPs/genes/miRNAs and phenotypes.Using GTExPortal (https://www.gtexportal.org/home/, accessed on 6 June 2023), miRNA TissueAtlas [79], Human Protein Atlas [80], and CNS microRNA Profiles database for mice [81], we investigated the expression of miRNAs and genes in different brain tissues of human and mouse samples.MiRNA target genes were extracted using miRTarBase (v.9) [82], miRDB (v.6) [83]), and TargetScan (v.8) [84].We used the over-representation analysis implemented in the miRNA Enrichment Analysis and Annotation Tool (miEAA 2.0) [85] to search for significant associations between sets of target miRNAs and disease outcomes, incorporating data from large miRNA, tissue, and pathway databases.Comparison with GWAS results: results for SNPs within target genes of significant miRNAs were looked up in publicly available GWAS summary statistics on white matter hyperintensity burden (dbGaP: phs002227.v1.p1 [11]).

## Figures and Tables

**Figure 1 ijms-25-00887-f001:**
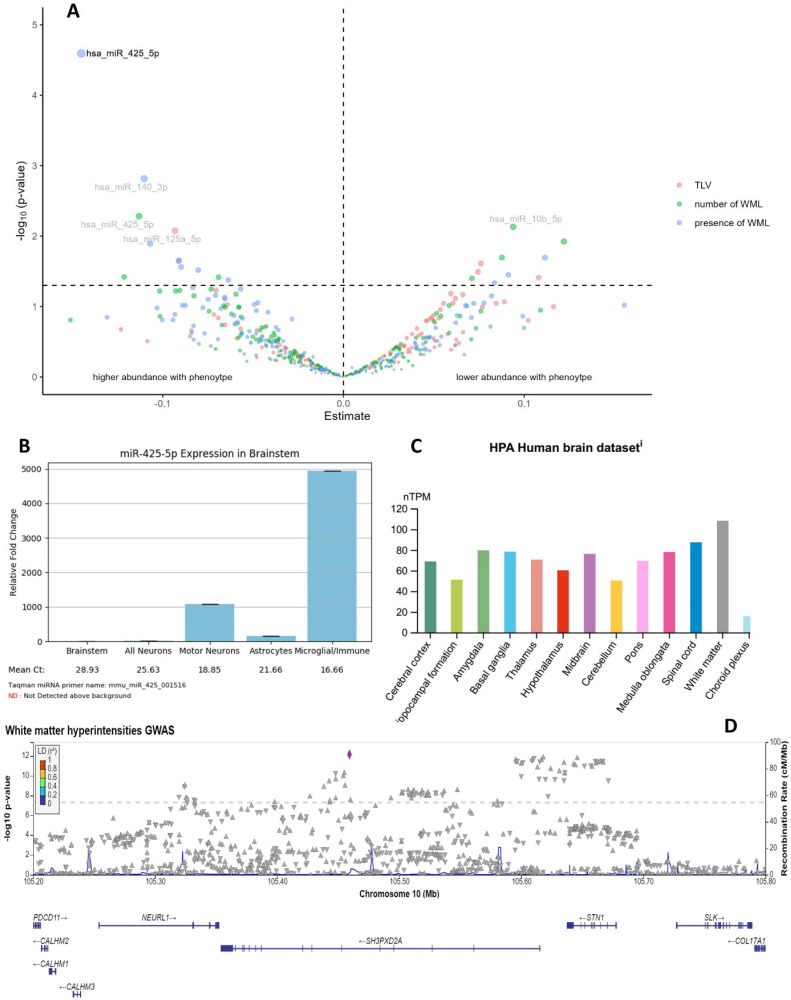
(**A**) Volcanolike−plot for the direct associations between 171 plasma-circulating miRNAs and WML parameters in the full miRNA/MRI sample (n = 648); the dashed horizontal line indicates nominal significance; only *hsa*-miR-425-5p (449 cases with WMLs and 189 controls without WMLs) remained significant after multiple testing correction showing a positive association towards the presence of white matter lesions. TLV: total lesion volume, WML: white matter lesions. Estimates refer to ΔCt values (ΔCt values > 0: lower abundance with phenotype, ΔCt values < 0: higher abundance with phenotype). (**B**) Expression of miR-425-5p in mouse brainstem in relation to other cell types. Expression is enriched in microglia cells (data from CNS microRNA profiles in mice; https://www.miRNA.wustl.edu, accessed on 4 October 2023). (**C**) Expression of *SH3PXD2A* gene across different brain tissues (nTPM = number of transcripts per million). Expression was highest in white matter (data from the Human Protein Atlas; https://www.proteinatlas.org/, accessed on 4 October 2023). (**D**) LocusZoom plot of the *SH3PXD2A* gene region (target gene of *hsa*-miR-425-5p) on chromosome 10 derived from the white matter hyperintensities GWAS including 50,970 individuals [11]. The direction of the triangle indicated positive (▲) and negative (▼) association with the endpoint.

**Table 1 ijms-25-00887-t001:** Sample characteristics of the TREND-0 analysis samples with available miRNA/MRI data and only with available MRI data (after QC and only including subjects with non-missing covariates).

	TREND-0 miRNA/MRI Sample * (n = 648)	TREND-0 MRI Sample (n = 1854)
**Sex**		
**Males**	328 (50.6%)	890 (48%)
**Females**	320 (49.4%)	964 (52%)
**Age in years**	50.3 (13.7), [21–79]	51.0 (13.9), [21–81]
**BMI**	27.2 (4.2), [17.7–48.0]	27.5 (4.4), [17.7–48.0]
**Systolic blood pressure (mmHg)**	124.9 (16.3), [88–196]	126.2 (17.2), [84–196]
**diastolic blood pressure (mmHg)**	76.5 (9.6), [51–115]	77.1 (9.9), [47–118]
**Hypertension**	251 (38.8%)	790 (42.7%)
**Current depressive symptoms (PHQ-9)**	12.7 (3.4), [9–35]	12.8 (3.5), [9–35]
**Education**		
**<10 years**	68 (10.5%)	273 (14.7%)
**=10 years**	369 (57%)	1011 (54.5%)
**>10 years**	211 (32.5%)	570 (30.8%)
**Smoking**		
**Never**	267 (41.2%)	732 (39.5%)
**Former**	247 (38.1%)	684 (36.9%)
**Current**	134 (20.7%)	438 (23.6%)
**Verbal memory immediate recall**	5.4 (1.2), [0–8]	5.4 (1.3), [0–8]
**Verbal memory delayed recall**	5.7 (1.6), [−3–8]	5.8 (1.7), [−3–8]
***APOE* ε4 carrier**	155 (24.0%)	448 (24.2%)
**ICV in cm^3^**	1563 (148), [1016–2040]	1560 (145), [1016–2040]
**WMLV in cm^3^**	0.56 (2.1), [0–27]	0.68 (2.4), [0–43.8]
**Number of WMLs**	3.0 (4.2), [0–37]	3.2 (4.4), [0–37]
**Presence of lesions**	454 (70.1%)	1320 (71.2%)

* TREND-0 miRNA/MRI sample is a subsample of the full TREND-0 MRI sample; PHQ: Patients Health Questionnaire; NA: not available; for metric variables, mean (sd) and range are given; for categorical variables, counts and percentages are given. WMLV: white matter lesion volume; ICV: intracranial volume.

**Table 2 ijms-25-00887-t002:** Overview of significant miRNAs in direct associations and interaction analyses in the TREND-0 miRNA/MRI sample (n = 648). Directions of effects (Pos/Neg), nominal *p*-values, and BH-corrected *p*-values in each analysis are listed and also the number of subjects for the specific miRNA analysis.

miRNA	WML Volume	Number of WMLs	Presence of WMLs	n
**Direct effects**				
***hsa*-miR-425-5p**	Pos, 0.46, 0.94	Pos, 0.005, 0.63	**Pos, 5.9 × 10^−5^, 0.01**	638
**Interaction with sex**				
***hsa*-miR-126-3p**	Neg, 1.7 × 10^−3^, 0.13	**Neg, 2.6 × 10^−4^, 0.044**	*-*	641
***hsa*-miR-374a-5p**	Neg, 0.03, 0.39	*Neg, 9.4 × 10^−4^, 0.08*	-	410
**Interaction with *APOE* ε4 carrier status**				
***hsa*-miR-140-5p ***	Neg, 0.024, 0.58	Neg, 0.001, 0.12	-	497
**Interaction with smoking status**				
***hsa*-miR-199a-5p**	**Neg, 1.0 × 10^−4^, 0.018**	**Neg, 2.6 × 10^−4^, 0.022**	-	516
***hsa*-miR-885-5p**	*Pos, 9.6 × 10^−4^, 0.083*	**Pos, 1.2 × 10^−4^, 0.022**	-	544
***hsa*-miR-194-5p**	Pos, 0.011, 0.32	**Pos, 7.8 × 10^−4^, 0.045**	-	619

Results adjusted for smoking, BMI, *APOE* ε4 carrier status, age, sex, batch, hematocrit (HCT), platelet count (PLT), educational attainment, hypertension, and ICV. Bold results are significant after Benjamini–Hochberg multiple testing correction (*p_BH_* < 0.05), italic results *p_BH_* < 0.1. * most significant miRNA with *p_BH_* = 0.12; interaction models were only tested for the dimensional end-points due to sample size. *Pos* denotes positive abundance with endpoint; *Neg* denotes negative abundance with endpoint. n: number of subjects. Estimates refer to ΔCt values (ΔCt values > 0: lower abundance with phenotype, ΔCt values < 0: higher abundance with phenotype).

**Table 3 ijms-25-00887-t003:** Significantly over-represented diseases using the six significant miRNAs as input. *p*-values corrected for multiple testing against 1330 diseases are displayed.

	miR-425-5p	miR-126-3p	miR-374a-5p	miR-199a-5p	miR-885-5p	miR-194-5p	FDR Corrected *p*-Value
**Neurodegenerative diseases**	x	x	x	x	x	x	0.005
**Niemann-pick disease**	x	x	x	x	-	-	3.5 × 10^−4^
**Alzheimer’s disease**	x	x	x	x	x	x	0.008
**Multiple sclerosis**	x	x	x	x	-	x	0.003
**Amyotrophic lateral sclerosis**	x	x	-	x	x	x	0.006
**Intellectual disability**	x	x	-	x	x	x	0.004
**Vascular disease**	x	-	x	x	x	x	0.018
**Brain disease**	x	x	-	x	x	x	0.012
**Huntington’s disease**	-	x	-	x	x	x	0.009
**Stroke**	-	x	x	x	-	x	0.011
**Schizophrenia**	-	x	-	x	x	-	0.011
**Hypertension**	x	x	-	x	-	-	0.023

miRNAs associated with the disease are marked with an x. All six miRNAs were associated with neurodegenerative diseases in general and with Alzheimer’s disease. “-” denotes no association with disease.

## Data Availability

Data from the SHIP study can be requested via the online application system and subsequent data transfer agreement (https://transfer.ship-med.uni-greifswald.de/FAIRequst/login?lang=en, accessed on 6 January 2024).

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
