# Peer review of "Circulating microRNA miR-425-5p Associated with Brain White Matter Lesions and Inflammatory Processes"

_ijms, 2024, doi:10.3390/ijms25020887_

Round 1
Reviewer 1 Report (New Reviewer)
Comments and Suggestions for Authors
The manuscript of Van der Auwera et al. investigate the association between 171 plasma-circulating miRNAs and White Matter Lesions (WML) in a cohort comprising 648 individuals from the general population. It was found to regulate various genes associated with WMLs with particular emphasis on the SH3PXD2A gene. Furthermore, miR-425-5p was found to be involved in immunological processes. In addition, noteworthy miRNAs associated with WMLs were identified, primarily moderated by the factors sex or smoking status. All identified miRNAs exhibited a strong over-representation in neurodegenerative and neurological disease. This manuscript is well written and presented. The study is well designed, and the methods are described in detail. The statistical methods and bioinformatics tools are appropriate. The results are very interestingly and innovative. The conclusions are clear and appropriate. However, the limitations of the study are not recognized by the authors. To demonstrate that miR-425 is a possible biomarker, functional studies are necessary. In addition, a new study involving patients and controls is necessary to confirm these results including analysis of ROC curves. I have no major correction for this excellent work.
Author Response
We thank the reviewer for this positive feedback. As suggested, we completed the list of study limitations emphazising the need for additional functional studies to prove miR-425 as a valuable biomarker for WML.
The limitation section now reads as follows: „To prove the hypothesis of has-miR-425-5p as a suitable biomarker for WML, additional clinical studies involving dementia patients. In addition, functional studies need to be performed to identify the implicated biological mechanisms of this target miRNA and derive regulatory models that could also drive drug discovery.“
Reviewer 2 Report (New Reviewer)
Comments and Suggestions for Authors
In the manuscript “Circulating microRNA miR-425-5p associated with brain white matter lesions and inflammatory processes” the authors analyzed plasma-circulating micro-RNAs (miRNAs) associated with WMLs. Doing so, in 648 individuals 171 plasma-circulating miRNAs were analysed and hsa-miR-425-5p as directly associated with WMLs was identified. 6 identified miRNAs exhibited a strong and significant over-representation in neurodegenerative and neurological diseases. The authors concluded that miR-425-5p holds the potential as a biomarker of WMLs shedding light on potential mechanisms and pathways in vascular dementia biomarkers, possibly for early detection of neurodegenerative processes leading to dementia. In the SHIP-study, data were correlated with respective MRI data showing WMLs.
The manuscript is well written and the results clearly presented. The statistics and methods are state of the art. The discussion also gives a good overview over the present knowledge on the respective 6 miRNAs. The discussion concentrates on the impact of the present manuscript with respect to the literature, especially it is worked out what is really new. Remarkably, the study is the first with a comprehensive association analysis between miRNA alterations and WMLs in a large general population sample, i.e., from the population-based Study of Health in Pomerania.
Author Response
We thank the reviwer for this positive feedback.
Reviewer 3 Report (New Reviewer)
Comments and Suggestions for Authors
In their paper entitled “Circulating microRNA miR-425-5p associated with brain white 2 matter lesions and inflammatory processes”, the Authors report that in a high number of patients with white matter lesions (WMLs), miR-425-5p was evidenced in the circulation; they also report on the various genes regulated by this miRNA, and, in particular, on the SH3PXD2A gene. Moreover, they found that many genes regulated by miR-425-5p are involved in immunological processes. Finally, on the basis of these observations, miR-425-5p is proposed as a biomarker for WMLs.
The paper is of interest and suitable for International Journal of Molecular Sciences. Moreover, the state of art, Materials and Methods, and Results are well described.
I only have two minor suggestions for the Authors: i) please, cite the complete name of the involved genes and not only the acronyms; ii) in order to allow the Readers to deal with the possible effects of an excess of the studied miRNAs, please, briefly explain the known functions of the genes probably regulated by them.
Comments on the Quality of English Language
English is fine, grammar should be checked.
Author Response
We thank the reviwer for this positive feedback. As suggested, we included the full names of the genes instead of only the acronymes when they first appear in the text. The second issue of the reviewer is difficult to handle as hundreds of genes can be regulated by one miRNA which cannot be described here. Nevertheless, we tried to briefly describe the main functions of the top WML associated genes regulated by the target miRNA has-miR-425-5p. These are SH3PXD2A, CTSS, EPN2, RIT1, and PPP4R3A. Additional functional studies will be needed to identify the target mechanisms and genes of miR-425-5p in the light of WMLs.
We included the following sentence: “These genes regulate pathways such as signaling pathways (Rho and RAC1 GTPase, p38 MAPKinase), immune system, neuronal pathways (neurotoxicity, neuronal development and differentiation), or cellular transport.”
This manuscript is a resubmission of an earlier submission. The following is a list of the peer review reports and author responses from that submission.
Round 1
Reviewer 1 Report
Comments and Suggestions for Authors
The manuscript entitled “Circulating microRNA miR-425-5p as potential regulator of brain white matter lesions through inflammatory processes” by Van der Auwera and colleagues describe the association of circulating levels of miR-425-5p with Brain white matter lesions and inflammation markers.
The manuscript compromises many data regarding different phenotypic and OMICs expression; however, a coherent story hardly emerges it has not enough quality to be published.
The manuscript is very speculative, authors only provide data from association and regression models with the data of human samples, they did not demonstrate the implication of this miRNA in these processes, in vitro and in vivo experiments with mimic or antagomir of this miRNA should be performed in order to prove this role as a regulator of brain white lesions and the implication of inflammatory mechanisms. Also, they mention possible target genes of this miRNAs but they did not demonstrate this regulation.
Furthermore, the manuscript is difficult to follow up, with too much data that are not relevant, such as a no significant effect of miRNAs with white matter volume in AD or measures related to. Also, a huge supplementary information that could be checked by visiting the web page that authors provide for each supplementary figure difficult the understanding and follow up of the manuscript. For example, figure S2 among others.
There is no cohesion in the number of samples in both subset of samples: In some cases, TREND-0 miRNA/MRI subset is composed by 648 samples, in others 647, and in other 708. The same occurs in TREND-0 MRI sample, authors said that is composed by 1854 samples and in other cases by 2047. This misunderstanding occurs again for the number of miRNAs analyzed, in some cases is 171 and in others 179.
Regarding that, why authors choose this 171-179 miRNAs, there is a big among of miRNAs expressed in plasma samples that they did not consider.
In supplementary figure S1A/B what represent red dots, blue dots, black dots? Please provide.
Author Response
Reply is given in the PDF

Reviewer 2 Report
Comments and Suggestions for Authors
The authors used 2 miRNA qPCR panels to study the SHIP subjects in regard to white matter lesions. However, it is not clear to me when the synthetic spike-in control was added, before storage, before RNA extraction, or before qPCR. This control is key to and critical for normalization between samples and thus needs to be emphasized. It will be better for the authors to clarify how normalization was done. Also, it is not clear to me how many controls and samples are in each category for each data analysis. The sampling size is also critical in statistic analysis. Other concerns are listed below:
1) Table 1. A justification on n=1854 from 2047 participants is needed, so as n=648 from 708, if any standard cutoff is used, which I cannot find in the manuscript.
2) Table 2. A justification is needed for n=647, different from n=648 in Table 1 and in manuscript. Moreover, the sample sizes for each control and each sample are needed, in addition to the total sample size. It will be better to list both p and pBH value in the same table for readers.
3) While Table 2 listed significant miRNAs, it will be better to list all miRNAs in Table S3, which helps readers easy to find the data, such as miR-152-3p, and reduces Table S1. Same to other Tables and supplementary tables.
4) Fig.1 sample size is needed in each figure if available, e.g., n for control and sample is needed for Fig.1A.
5) Table 3. It will be better to include miR140-5p as well, since it has been shown in both Tables 2 and 4.
Comments on the Quality of English LanguageN/A
Author Response
Reply is given in the PDF

Round 2
Reviewer 1 Report
Comments and Suggestions for Authors
The new manuscript entitled “Circulating microRNA miR-425-5p as potential regulator of brain white matter lesions through inflammatory processes” by Van der Auwera and colleagues assess almost all the minor issues proposed. However, the manuscript still very speculative, just association and regression models that links WML with miR-425-5p is not sufficient to said that this miRNA is a regulator of WML formation and neither that this regulation is through inflammatory processes. It is reasonable thing and results showed by authors are in line with a possible role of this miRNA in WML formation and a possible implication of inflammatory mechanisms in the process, but it is necessary additional experiments in vitro and in vivo to confirm that.
Authors said that they are aware that Additional experiments are needed to identify the implicated biological mechanisms, but they have not expertise in this field and is not feasible with their data. In this case, maybe authors should thing if they should reformulate their study and check if this miRNA could be a good biomarker to do an early diagnostic of the diseases related to WML or make a review/meta-analysis of literature and include their data or they should thing in submit the manuscript to another journal with less impact factor.
Reviewer 2 Report
Comments and Suggestions for Authors
N/A